Biomarker potential of repetitive-element transcriptome in lung cancer

Arroyo Macarena 1 2
Bautista Rocío 3
Larrosa Rafael 3 4
Cobo Manuel Ángel 5
Claros M. Gonzalo claros@uma.es 2 3 5
1 U.G.C. Médico-Quirúrgica de Enfermedades Respiratorias, Hospital Regional Universitario de Málaga , Málaga , Spain
2 Department of Molecular Biology and Biochemistry, Universidad de Málaga , Málaga , Spain
3 Andalusian Platform for Bioinformatics-SCBI, Universidad de Málaga , Málaga , Spain
4 Department of Computer Architecture, Universidad de Málaga , Málaga , Spain
5 Area of Oncology and Rare Diseases (IBIMA), Hospital Regional Universitario de Málaga , Málaga , Spain
Uversky Vladimir
Electronic publication date: 2019 Dec 19
Publication date: 2019
Volume: 7
Electronic Location ID: e8277
Received 2019 Oct 2; Accepted 2019 Nov 22
Copyright: ©2019 Arroyo et al.
Copyright year: 2019
Copyright holder: Arroyo et al.
License: This is an open access article distributed under the terms of the Creative Commons Attribution License, which permits unrestricted use, distribution, reproduction and adaptation in any medium and for any purpose provided that it is properly attributed. For attribution, the original author(s), title, publication source (PeerJ) and either DOI or URL of the article must be cited.
License URL: https://creativecommons.org/licenses/by/4.0/

Keywords: Repetitive element, Lung cancer, Differential expression, Biomarker, Transcriptome

Funding: Neumosur 12/2015, 14/2016 and 5/2017 European Union through the European Regional Development Fund (ERDF) 2014-2020 Spanish Agencia Estatal de Investigación TIN2017-88728-C2-1-R This work was supported by Neumosur (grants 12/2015, 14/2016 and 5/2017), and co-funded by the European Union through the European Regional Development Fund (ERDF) 2014-2020 “Programa Operativo de Crecimiento Inteligente” together with Spanish Agencia Estatal de Investigación (TIN2017-88728-C2-1-R). The funders had no role in study design, data collection and analysis, decision to publish, or preparation of the manuscript.

==============================
Since repetitive elements (REs) account for nearly 53% of the human genome, profiling its transcription after an oncogenic change might help in the search for new biomarkers. Lung cancer was selected as target since it is the most frequent cause of cancer death. A bioinformatic workflow based on well-established bioinformatic tools (such as RepEnrich, RepBase, SAMTools, edgeR and DESeq2) has been developed to identify differentially expressed RNAs from REs. It was trained and tested with public RNA-seq data from matched sequencing of tumour and healthy lung tissues from the same patient to reveal differential expression within the RE transcriptome. Healthy lung tissues express a specific set of REs whose expression, after an oncogenic process, is strictly and specifically changed. Discrete sets of differentially expressed REs were found for lung adenocarcinoma, for small-cell lung cancer, and for both cancers. Differential expression affects more HERV-than LINE-derived REs and seems biased towards down-regulation in cancer cells. REs behaving consistently in all patients were tested in a different patient cohort to validate the proposed biomarkers. Down-regulation of AluYg6 and LTR18B was confirmed as potential lung cancer biomarkers, while up-regulation of HERVK11D-Int is specific for lung adenocarcinoma and up-regulation of UCON88 is specific for small cell lung cancer. Hence, the study of RE transcriptome might be considered another research target in cancer, making REs a promising source of lung cancer biomarkers.

Introduction

Lung cancer is the most frequent cause of cancer death worldwide, with a five-year survival rate of about 16% (Siegel, Miller & Jemal, 2016). Reasons for these poor results include late diagnosis, advanced disease stage at time of presentation, and limited therapeutic options (Stewart & Wild, 2014). Most lung cancer patients are treated with first and second lines of chemotherapy, while only a small fraction (5–7%) are candidates for targeted therapies, even though this approach is challenged by poor tumour-targeting, off-target effects, and development of resistance to therapy (Srivastava et al., 2018). Spurred on by these challenges, new biomarkers that can be assayed with minimal patient burden are always welcome. Interestingly, it is well demonstrated that RNA-seq could improve medical care (Karam et al., 2019) and merits its clinical applicability (Byron et al., 2016).

The oncogenic change is considered a cellular reprogramming orchestrated by the up- or down-regulation of a series of protein-coding genes. To date, those genes constitute the main source of genomic biomarkers, even though they account for 1.5% of the genome (ENCODE Project Consortium, 2012). This is the case of CBX3 and CRABP2 (Han et al., 2014) and other genes (Tian et al., 2017), the long non-coding RNA Veluct (Seiler et al., 2017), and more and more microRNAs (Daugaard et al., 2017). Moreover, several lung cancer signatures based on expression of genes and non-coding RNAs have been proposed (Shukla et al., 2017; Girard et al., 2016). But it is an often overlooked fact that nearly 53% of the human genome corresponds to repetitive sequences and that there are many cell RNAs besides protein coding mRNAs and miRNAs usually studied as biomarkers. Up to 85% of the repetitive genome are transposons or transposon-derived sequences (De Koning et al., 2011) that can produce nowadays neglected RNAs with respect to their biomarker capability (Gnanakkan et al., 2013).

Transposons are classified as class I and class II (or DNA transposons). Class I transposons with a retroviral origin (Suntsova et al., 2015) have flanking LTRs (Long Terminal Repeats) that can act on neighbour genes. Human endogenous retroviruses (HERVs) belong to this family and are related to genetic diseases, including cancer. Incomplete transpositions, recombination and the absence of selection pressure on transposons can result in the presence of mutated, solitary LTRs or other parts of them (Vargiu et al., 2016). The most abundant RNA transposons in the human genome are LINEs (Long INterspersed Elements; about 20%) and SINEs (Short INterspersed Elements; about 13%) (Ayarpadikannan & Kim, 2014). LINE-1, which only has 100 active copies per human genome (Xiao-Jie et al., 2016), can serve as a prognostic factor of cancer progress, and even as a therapeutic target (Apostolou et al., 2015). In contrast, SINEs are non-autonomous since they do not encode proteins and require LINE-coded proteins for their propagation. The most successful SINEs found in any organism are Alu sequences, with about 1.1 million copies, although most of them are not transcriptionally active (see Ade, Roy-Engel & Deininger, 2013 for a review). They are classified into several subfamilies, Alu J, S, and Y, from the oldest to the youngest, where only younger variants (about 150 copies) can move throughout the genome (Wang & Huang, 2014). Alu sequences, particularly those found in introns on both the 5′-end and 3′-end, were found embedded in short and long ncRNAs, where they are found to directly participate in base-pairing to mRNA targets. Reports on Alu transcription as ncRNAs indicate that they seem to be critical players in gene regulation, alternative splicing, alternative polyadenylation and molecular pathways (Chen & Yang, 2017). Therefore, the transcription of transposons, transposon-derived elements, and other REs, would result in or from changes in the cell, for example oncogenic changes.

The repetitive elements (REs) that conform the repetitive genome are gathered in RepBase (Kojima, 2018), one of the most comprehensive databases including the classification of eukaryotic repetitive sequences (Kojima, 2018; Bao, Kojima & Kohany, 2015), even though some repetitions among ‘Eutr’ (eutherian transposon), ‘EUTREP’ (eutherian repeat), ‘UCON’ (ultraconserved element), and ‘Eulor’ (euteleostomi conserved low frequency repeat) are harder to classify (Kojima, 2018). Some REs can contain regulatory sequences (acting as promoters and transcription signals) that enable them to dysregulate adjacent genes and drive to, for example, cancer (Pavlicev et al., 2015; Trizzino, Kapusta & Brown, 2018; Jang et al., 2019). A body of evidence is accumulating against the non-specific dysregulation of RE and favouring the concept of a fine-tuned change of expression after an oncogenic process (Gibb et al., 2015; Clayton et al., 2016; Larrosa et al., 2018). That could explain why ‘read-through’ transcription of intronic transposons and interspersed repeats occurs, and why this expression is tissue- or disease-specific (Gnanakkan et al., 2013). In fact, it has been demonstrated that most RE RNAs (about 99% in the case of LINE-1 Deininger et al., 2017) arise from read-through transcription due to their occurrence in introns or ncRNAs. In fact, RE RNAs were found in normal and diseased tissues in experimental models and in humans (O’Donnell, Burns & Boeke, 2008; Kaneko et al., 2011; Lee et al., 2012). Therefore, the TE transcriptome (sometimes referred as transposon transcription in above cited literature) has been studied in different contexts, indicating that it is affordable and produces consistent results.

As already proposed by Faulkner et al. (2009), expression profiling of REs might serve to characterise pathologic states including cancer since RE expression might generate, among other things, tumour-specific antigens or chimeric transcripts. Reports proving their theranostic marker capability are available (Sacha et al., 2012). Bioinformatic tools such as RepEnrich (Criscione et al., 2014) have been developed to quantify differential expression of REs, demonstrating, for example, that LINE-1 is up-regulated in prostate cancer cells (Criscione et al., 2014) and HERVs are down-regulated in psoriasis (Lättekivi et al., 2018). Hence, discovery of new lung cancer biomarkers based on RE transcriptome was faced since it is feasible and underdeveloped. This required the implementation of a bioinformatic workflow to study the RE expression change in two lung cancers (lung adenocarcinoma (LUAD) and small cell lung cancer (SCLC)) in matched healthy-tumour cells from the same patient to show that more HERV- than LINE-derived REs specifically change their expression in lung cancer. In particular, two differentially down-regulated REs (AluYg6 and LTR18B) were proposed as consistent biomarker candidates for both LUAD and SCLC diagnosis, another two (UCON88 and HERVK11D-Int) are differentially up-regulated, and a further 33 were proposed as SCLC-specific diagnostic biomarkers.

Methods

Sample selection

Major genomic databases were searched for RNA-Seq samples of lung cancers that met the following criteria: (1) reads obtained only from cryogenically frozen tissues using a preservation protocol ensuring RNA integrity, even if paraffin-fixed tissues have been recently described as suitable for RNA-seq (Bossel Ben-Moshe et al., 2018); (2) normal and tumour tissues must be derived from the same patient; and (3) sequencing of total RNA from normal and tumour tissues was performed in high-throughput platforms. Only two datasets were found: Bioproject EGAS00001000334 from the EGA database containing 2 × 75 bp paired-reads from 17 patients with SCLC (Rudin et al., 2012) generated with an Illumina HiSeq2000, and Bioproject ERP001058 from NCBI databases containing 2 × 100 bp paired-reads from 50 patients with LUAD (Ren et al., 2012) also generated in a HiSeq2000. Usage authorisation requirements were fulfilled.

As a validation cohort, eight suitable frozen biopsies of LUAD were obtained from Biobanco del Sistema Sanitario Público de Andalucía (http://www.juntadeandalucia.es/salud/biobanco/). They correspond to patients from Malaga Regional Hospital (MRH) in Spain. Total RNA from the normal and the cancerous tissue was extracted as described in Arroyo et al. (2018). Libraries were stranded-first sequenced to produce 2 × 75 bp reads in the NextSeq 550 of the University of Málaga, and deposited as Bioproject PRJNA563806. The Ethics Committee of the Regional Hospital of Málaga called “Comité de Ética de la Investigación (CEI) Provincial de Málaga” (CIF Q-9150013-B) approved and consented this study on 28/01/2016. The consent form was obtained when the samples were resected and frozen, during 2011.

Architecture, software and databases

The bioinformatic workflow has been tested, implemented and executed on a SUSE Linux Enterprise Server 11SP2 with Slurm queue system and Infiniband network (54/40 Gbps) consisting of 216 nodes with Intel E5-2670 2.6 GHz cores for a total of 3,456 cores and 8.4 TB of RAM. It requires installation of the following tools: SeqTrimNext (Falgueras et al., 2010) for dataset cleaning, RepEnrich (Criscione et al., 2014) as the key tool for the RE expression analysis, Bowtie1 (Langmead, 2010) for mapping, SAMtools (Li et al., 2009) for the manipulation of the mappings, and the R packages edgeR (Robinson, McCarthy & Smyth, 2010) and DESeq2 (Love, Huber & Anders, 2014) for differential expression analyses and graphical representations. Venn diagrams were obtained at http://bioinformatics.psb.ugent.be/webtools/Venn.

Human genome version hg38.p1 (GCA_000001405.16) was used as reference for mapping. Following the instructions of Criscione et al. (2014), it was formatted as necessary and called Hg38_RE in the present study. RepBase (Bao, Kojima & Kohany, 2015) was processed to produce hRB_RE containing 1,267 human REs sequences spreading over 4,713,583 positions of the genome. RepBase classification of REs according to families (types), classes and individual elements was preserved.

Read pre-processing

The automated workflow designed in this study can be divided into three main blocks: data pre-processing, quantification and differential expression (Fig. 1). Contaminating sequences, adaptors, low quality regions, etc., were removed from raw reads (Raw reads in Fig. 1) using SeqTrimNext applying the standard parameters for Illumina paired reads. The resulting clean reads from every SAMPLE were mapped to the reference human genome using Bowtie1 (the version required by RepEnrich), and then sorted and indexed using SAMtools, to obtain one sorted BAM file and one FastQ file containing the multi-mapping reads for each SAMPLE (Fig. 1).

Figure 1 Schematic workflow for assessing differential expression of REs from matched-sample raw reads developed in this study.

Each main block is explained in detail in the main text. Grey rectangles indicate bioinformatic tools and filters; lilac boxes correspond to R functions; green and yellow boxes are relevant input or output data to be saved; and dark green cylinders refer to databases.

RE quantifications

In the second block of the workflow (Fig. 1), quantification of RE expression was obtained using RepEnrich with the following files obtained in the previous step: (1) FastQ files containing paired-reads mapping more than once on the human genome, (2) sorted, indexed BAM files with reads mapping only once on the human genome, (3) the human genome sequences in the appropriate format (Hg38_RE), and (4) the RepBase human subset of REs in the appropriate format (hRB_RE). As a result, three files per sample were obtained: one with the expression of each RE, another with the expression level for each RE class, and another for each RE family. Throughout this study, only data from RE file were used, since data from classes and families were not relevant (Tables S3 and S6).

All snRNA, rRNA and tRNA elements were removed from output because they can provide spurious differences due to the way each laboratory extracted the RNAs. This resulted in 1,190 analysed REs. REs with low expression level (less than 10 counts per million in at least 2/3 of the samples) were also discarded. Finally, filtered RE expression data for all samples were gathered in a single matrix y (Fig. 1).

Differential expression

The expression matrix y for each SAMPLE was analysed in two parallel ways: (1) to determine the variability of RE expression change (calculated as a binary logarithm of fold-change, logFC) within a set of patients, a logFC per Patient (logFCpP) was calculated (left arm in last block of Fig. 1); and (2) an overall estimation of the logFC (logFCOE) was determined to perform the statistical analyses that drive to differentially expressed REs (right arm in last block of Fig. 1). Since both values are expressed as binary logarithms of matched samples, up-regulation and down-regulation provide symmetric values of around zero and were comparable.

The logFCpP was based on the cmp function of edgeR package taking into account that samples were matched by patient (normal/tumour). The obtained matrix logcpm contained the normalised expression of each RE on the rows, with a SAMPLE in each column. Finally, logFCpP was calculated, per patient, as follows:

lfc ←logcpmtumor − logcpmnormal

where rows in the lfc matrix contained logFCpP for each RE, and columns corresponded to patients.

The logFCOE was necessary to determine whether the expression change of a RE was statistically significant or not. Data contained in y were conveniently processed using function DESeq from the DESeq2 package taking into account the normal/tumour samples belonging to the same patient to obtain the object dds that contained the logFCOE values and their significance together with, among other data. An expression change was considered statistically significant when |logFCOE| > 1 with an FDR < 0, 05.

The workflow finally provided graphical representations of the analysed data, stored in several PDFs for convenience (bottom of Fig. 1).

Results

Sample quality controls

First, libraries from LUAD (ERP001058) and SCLC (EGAS00001000334) were plotted based on the premise that similar samples should form a cluster (Robinson, McCarthy & Smyth, 2010). While LUAD samples provided two distinct groups (tumour and normal), SCLC contained one normal sample (S585275) mixed with tumour samples (Fig. 2). Since misclassification could be assigned to some kind of mishandling during RNA preparation, this sample was discarded together with its matched tumour sample S585270. Therefore, samples from the 50 LUAD patients and only 16 SCLC patients were then pre-processed.

Figure 2 SCLC sample clustering based on gene variation between replicate RNA samples as returned by plotMDS function from edgeR package.

Normal lung samples (black) form a homogeneous and distinct cluster clearly separate from the tumour samples (red) in the first dimension; only the sample encircled in purple is misclassified. BCV, Biological Coefficient of Variation.

The overall pre-processing results of samples grouped by cancer types is shown in Table 1, where it can be observed that only 12.29% of LUAD reads and 7.29% of SCLC reads were discarded. The level of multi-mapping reads (corresponding to REs and other repetitive sequences) is only 6.91% and 6.32% in LUAD and SCLC, respectively. The reasonable number of biological replicates, the high number of useful reads, the low rate of rejection, and a reasonable rate of multi-mapping reads ensured that results derived from these data could be representative of lung cancer.

Since samples from LUAD and SCLC were obtained from different laboratories (LUAD: Macrogen, Korea; SCLC: Genentech, CA, USA), from different sources (LUAD: Seoul St. Mary’s Hospital, Korea; SCLC: Johns Hopkins tissue repository, USA), using different RNA preparation protocols (LUAD: RNAiso Plus Kit from Takara Bio Inc.; SCLC: Qiagen AllPrep RNA kit), the RE expression was investigated in normal lung cell samples to reveal whether any differential behaviour among both samples could be detected. Surprisingly, 30 differentially expressed REs were found (Table S1), in which most (25) were up-regulated in normal cells from patients with SCLC, one was clearly (−1.39), and other 4 were slightly (−1.063 to −1.16), up-regulated in normal cells from LUAD samples. This suggested that patients and/or laboratories and/or RNA preparation protocols were influencing the sequencing result. Moreover, this showed that the only way to compare these SCLC and LUAD samples disregarding any bias was to match the logFC values per patient for each RE, as in the calculation of logFCpP described in Methods.

Table 1 Summary of read pre-processing and mapping against hg38.p1 grouped by lung cancer type.

Lung	# of	# of total	# of useful	Rejected	Multi-mapping	
cancer	patients	reads	reads	reads (%)	reads (%)	
LUAD	50	87,265,713	71,999,479	12.29	6.91	
SCLC	16	77,399,467	68,298,400	7.29	6.32	

Overall RE expression after oncogenic reprogramming was also analysed per patient in each lung cancer. It revealed that the median values of logFCpP for all REs was statistically equal to zero in both LUAD and SCLC Fig. S1. This demonstrated that there was no broad RE deregulation after an oncogenic change in lung cells, which led to the analysis of overall RE differential expression by disease, not by patient. This is also supported by the presence of outliers in Fig. S1.

Differentially expressed REs in LUAD

The 50 LUAD patients provided 100 matched samples that served to identify 15 differentially expressed REs, most of them showing very significant fold changes (1.1 to 2.0 and −1.1 to −2.1, FDR < 10−8; Fig. 3 and Table S2). The advantage of this kind of plot is that it provides an overview of the logFCOE used for differential expression (triangles in Fig. 3), together with the distribution of fold-change values for each patient based on the logFCpP (boxes and whiskers in Fig. 3). As a sign or reliability, each median logFCpP and its corresponding logFCOE were consistent in spite of being calculated by different methods, supporting the consistency of the comparison and allowing to discern REs whose expression was consistent throughout patients (AluYg6 and MER126 among others), and those presenting a high variability, notably MER65-Int.

Figure 3 Expression distribution of the 15 differentially expressed REs in LUAD patients. Alphabetically ordered names are at the x axis, and the y axis corresponds to the binary logarithm of expression fold-change (logFC).

Boxes and whiskers correspond to logFCpP values, where black dots are outliers, while triangles denote the logFCOE (see Table S2). Note that all triangles (representing logFCOE) are within the interquartile rage (IQR) of logFCpP values, usually close to median, indicating that the single measure considering all patients (logFCOE) is consistent with the distribution of RE expression changes for all patients (logFCpP), but is less informative.

Of the 15 differentially expressed REs, seven (HERVL18-int, LTR4, LTR18A, ALR_Alpha, HERVK11D-int, MER65-int and HERV3-int) were up-regulated and eight (UCON80, LTR77, L1MEa, UCON34, MER126, UCON26, AluYg6 and LTR18B) down-regulated in LUAD cells with respect to normal lung. Two REs (MER126 and UCON80) were derived from DNA transposons, 8 REs were endogenous retroviruses (ERVs) of LTR class, and the last was a L1. Surprisingly, down-regulation of L1MEa was observed in Fig. 3, in spite of this RE being widely reported as up-regulated in many cancers (Criscione et al., 2014; Ponomaryova et al., 2017). Finally, one centromeric satellite (ALR_alpha) was up-regulated while one SINE (AluYg6) and two REs of unknown type were down-regulated. When considering differential expression by RE class or family in LUAD, only the centr family is slightly differentially expressed (logFC = 1.02; Table S3). In conclusion, it seems that ERVs of the LTR class were the most affected REs after LUAD-specific expression changes.

Differentially expressed REs in SCLC

The 16 patient samples of SCLC which met our criteria provided 32 matched samples whose analysis revealed 71 differentially expressed REs (Fig. 4), all of them with FDR < 10−4 (Tables S4 and S5). As occurred in LUAD, logFCpP medians and logFCOE values were also consistent in SCLC (Fig. 4). Many REs displayed small boxes (indicating a highly coincident expression change), while others, such as HERVL18-Int, MER65-Int and MLT2B5, presented a more widespread distribution of logFCpP values. Half of the differentially expressed REs (36) were ERVs of the LTR class, 15 were derived from DNA transposons, 6 were SINEs, 5 were LINEs (L1MEa, L1M2a1, CR1_Mam, L1M3a and L1PA12), 3 were satellites and 5 were of unknown class (Tables S4 and S5). Although more LINEs (5) appeared in this type of lung cancer compared to LUAD, the number of ERVs was again comparatively higher (36). In this disease, the bias towards down-regulation was more apparent: 50 differentially expressed REs were down-regulated against 21 differentially expressed REs up-regulated. Also consistent with LUAD, differential expression by RE class or family in SCLC is minimally observed only for Satellite class, as well as telo, MIR, centr and hAT families (marked with asterisks in Table S6).

Figure 4 Expression distribution of the 71 differentially expressed REs in SCLC patients following the same plotting criteria described in Fig. 3.

Details of logFCOE (triangles) are provided at Tables S4 and S5. Note that box medians and triangles are again in close proximity.

To show the down-regulation bias, the 71 differentially expressed REs were grouped by their RepBase class and then plotted (Fig. 5). This representation confirmed that the median logFCpP of all classes fell in the down-regulated part of the plot, with the exception of satellites, that were clearly up-regulated. In the case of LINEs, even if three were down-regulated (L1MEa, L1M2a1, CR1_Mam) and only two up-regulated (L1M3a, L1PA12), the median logFCpP was clearly on the down-regulation side, and the same was true for more populated classes such as LTR and DNA. Hence, even if particular REs were up-regulated and others down-regulated, the overall picture in SCLC samples was repression of REs when lung cells were compromised by oncogenic changes.

Figure 5 Expression distribution of the 71 differentially expressed REs from SCLC of Fig. 4 grouped by their class in RepBase.

DNA? class box is so neat because it contains only one RE belonging to this class: UCON14 (see Tables S4 and S5).

Potential lung cancer biomarkers

Differentially expressed REs for SCLC (Fig. 4), LUAD (Fig. 3) and normal lung (Table S1) were compared to obtain REs that could be disease-specific and those shared by diseases (Fig. 6). Most differentially expressed REs (79 = 55 + 24 in Fig. 6) were specific for normal lung and SCLC, while only four were specific for LUAD. Surprisingly, 6 REs from SCLC (6 = 5 + 1 in Fig. 6) overlapped with those of normal lung cells, and one (MER65-int) appeared in the three populations. Interestingly, Fig. 6 also revealed that 10 REs (namely LTR77, L1MEa, UCON26, HERVL18-int, UCON34, LTR18B, AluYg6, UCON80, MER126 and LTR18A) were shared by both lung cancer samples.

Figure 6 Distribution of the 99 unique REs that were differentially expressed in normal lung, LUAD and/or SCLC.

The 10 common REs and MER65-Int were selected in the first instance as the “primary set” of biomarker candidates and their expression changes were extracted in Fig. 7A for convenient comparison. It became evident that MER65-int, the RE common to the three samples in Fig. 6, presented statistically significant values of logFCOE in LUAD and SCLC (FDR < 10−4), but the logFCpP values were highly dispersed in both populations, making it useless as a biomarker. Since logFCOE values and boxes-and-whiskers of logFCpP of this primary set display the same tendency in both lung cancers, there might be a common reason for such a change. Once MER65-Int was excluded, the comparable expression change of the 10 remaining biomarker candidates (AluYg6, HERVL18-int, L1MEa, LTR18A, LTR18B, LTR77, MER126, UCON26, UCON34 and UCON80) made them a “suitable set” of biomarker candidates for both lung cancers.

Figure 7 Expression distribution of REs selected as potential biomarkers for LUAD and SCLC, plotted using the same criteria as described in Fig. 3.

(A) The 11 REs of the primary set of biomarker candidates in LUAD and SCLC. (B) The 10 REs of the suitable set (namely, those of A without MER65-Int) of biomarker candidates in the seven new patients with LUAD studied in Arroyo et al. (2018).

Ideal confirmation of the suitable set of biomarker candidates requires a new cohort of patients. Since it is not easy to find a public lung cancer dataset using matched total RNA sequencing samples where RE expression can be calculated, seven suitable patients with LUAD from Malaga Regional Hospital (MRH) in Spain studied in Arroyo et al. (2018) were used. Unfortunately, no samples for SCLC were found. Results in Fig. 7B confirmed that all REs behave as expected in this new cohort and supported the hypothesis of a consistent and specific expression change of these REs due to lung cancer.

Our proposal is that potential biomarkers for research or clinical use should preferably present the same expression change (up-regulation or down-regulation) in all analysed patients, rather than in most patients. Hence, biomarker candidates shown in Fig. 7 were filtered on the basis of their IQR as follows: no candidate should include logFC = 0 within its expression fold-change range defined by its median logFCpP ± 1.5 × IQR (the box-and-whiskers in Fig. 7). Most differentially expressed REs in Fig. 7A contained the logFC = 0 within the whiskers in at least one of the diseases, indicating that, even if they presented consistent values for logFCpP and logFCOE in lung cancers, there were individual patients whose RE-specific expression change occurred in the opposite direction to the majority. Fortunately, three exceptions —AluYg6, LTR18B and MER126—presenting logFC = 0 outside their ±1.5 × IQR range were found, suggesting that they could be potential biomarkers for lung cancer in clinical practice. In the search for confirmation with the 7 new LUAD patients in Fig. 7B, MER126 presented a narrow logFCpP range that included logFC = 0, thus precluding it from being a clear biomarker; only AluYg6 and LTR18B could be proposed as consistent biomarkers for both LUAD and SCLC.

Potential biomarkers for SCLC or LUAD

The same filtering criteria described above were applied to SCLC-specific and LUAD-specific differentially-expressed REs in Fig. 6. The four LUAD-specific REs (ALR_Alpha, HERV3-Int, HERVK11D-Int and LTR4) were all up-regulated in tumour cells, but the logFC = 0 value fell within their logFCpP ± 1.5 × IQR range (Fig. 3). Moreover, the Venn diagram in Fig. 8 (left) confirmed that three were also expressed—although not differentially expressed—in SCLC and normal lung cells, while only one (HERVK11D-Int, marked with a bullet (•) in Table S2) seems to be expressed in LUAD from the MRH, but not SCLC. It can be hypothesised that HERVK11D-Int expression may be specifically up-regulated in LUAD and not SCLC.

Figure 8 Lung cancer-specific REs distributed by disease and dataset.

(A) matches of LUAD-specific REs with all REs expressed in SCLC from databases (DB) and LUAD from the Malaga Regional Hospital (MRH). (B) matches of SCLC-specific REs from SCLC with all REs expressed in LUAD from DB and from MRH.

With regard to the 55 SCLC-specific REs in Fig. 6, 33 did not include the logFC = 0 value within their logFCpP ± 1.5 × IQR range. In fact, 26 out of the 33 were down-regulated (marked with an asterisk in Table S4), while 7 were up-regulated (marked with an asterisk in Table S5). Interestingly, the Venn diagram of Fig. 8 (right) shows that among the 55 SCLC-specific REs, only UCON88 (marked with a bullet (•) in Table S5) is not expressed at all in any LUAD cells but is up-regulated in SCLC, suggesting that it is a good candidate for SCLC-specific biomarker. Unfortunately, REP522 (already described as a putative biomarker for cancer in Kaczkowski et al. (2016) and patent application WO2012/031008A2 to Skog et al. (2012)) is expressed in some LUAD patients from MRH, making it a less clear biomarker than UCON88.

Discussion

Bioinformatic workflow on SCLC and LUAD samples

Sequenced datasets from frozen tissues in Bioprojects EGAS00001000334 for SCLC and ERP001058 for LUAD (Table 1) provided good quality control analyses, despite the removal of one patient from the SCLC dataset (Fig. 2). This is illustrated by the reasonable number of matched biological replicates and the high number of reads per sample longer than 40 nt that remained useful. The number of multi-mapping reads (6.91% in LUAD and 6.32% in SCLC, Table 1) is lower than described for prostate cancer (12.43% by (Criscione et al., 2014), 12.44% using our workflow) and mouse lung cancers (Faulkner et al., 2009), or the 27.5% reported in psoriatic skin (Lättekivi et al., 2018). The underlying cause of the low level of multi-mapping reads in SCLC and LUAD samples might be more likely related to biological (lower RE expression) or methodological (removal of RE sequences during RNA preparation) reasons rather than bioinformatic processing (more stringent read pre-processing, different mapping functions and filtering).

Quality control and the further mapping and expression analyses were fully automated using established bioinformatic tools, integrating autonomous applications with R packages. The workflow was designed to treat matched-samples, but it can also analyse unmatched samples only by changing the experimental design file. As a novelty, it provides two estimates of the expression change for REs. The first estimate, logFCpP, illustrates the distribution of expression changes for every RE per patient and allows the filtering off of any candidate whose range of logFCpP values (namely, the box-and-whisker interval in boxplots) includes logFC = 0. The second estimate, logFCOE, is the fold-change estimate that served to calculate the differential expression. Both logFCpP and logFCOE were plotted together (for example, Figs. 3 and 4) to reveal that logFCOE per se is unsatisfactory for biomarker selection. For example, in the case of MER65-Int in Fig. 7A, it presents statistically significant differences (FDR < 10−4) of logFCOE in LUAD and SCLC, but its wide range of expression (logFCpP) in both cancers spans logFC = 0, precluding its qualification as a biomarker candidate.

Not all healthy lung cells express the same REs

Mammalian lung cells are known to express a detectable level of REs (Faulkner et al., 2009). Potential population differences were obtained when normal lung samples from LUAD and SCLC patients were compared (Table S1). Unexpectedly, healthy (normal) lung samples from LUAD and SCLC patients presented 30 differentially expressed REs, with 25 up-regulated in healthy lung cells from SCLC and only 5 up-regulated in healthy lung cells from LUAD patients. Both technical reasons (laboratory, extraction kit, sequencing platform, etc.) and population differences (genetic background, local environment, lifestyle, cancer management, time between presentation and surgery, etc.) may explain the differences. Since it has been demonstrated that cigarette smoking alters the transcription level of many genes in normal lung cells of LUAD, although only seven genes up-regulated in all studies (Pintarelli et al., 2019), our results can extend the same conclusion to the RE transcriptome. Irrespective of the rationale for these differences, the 30 differentially expressed REs in normal cells cannot be considered lung biomarkers. In fact, six were also found in lung cancer cells (Fig. 6), indicating that their expression change is not driven by oncogenic signals. Far from being a flaw, differentially expressed REs in different healthy lungs strongly increase the significance of common REs in different lung cancers, and suggest that matched normal-tumour sample sequencing is recommended to guarantee a reliable differential expression. Additionally, these results are in agreement with those reported for L1, where only a very restricted subset of loci significantly contributes to the bulk of L1 expression, governed by individual-, locus-, and cell-type-specific determinants (Philippe et al., 2016).

HERV-derived RE down-regulation in lung cancer

Overall RE expression level does not vary after a lung oncogenic change (Fig. S1) and only a ‘small’ number (13 and 71 in Figs. 3 and 4, respectively) of REs are differentially expressed. Differential expression by RE class was undetectable, and only a few families (Satellite, telo, MIR, centr and hAT) show minimal differential expression (Tables S3 and S6), indicating that RE expression change should not be related to families but particular REs. It further supports the established proposal that RE expression seems to be strictly controlled before and after any oncogenic transformation (Gibb et al., 2015; Clayton et al., 2016; Lättekivi et al., 2018; Arroyo et al., 2018) and cannot be explained by a broad deregulation as initially proposed by Shalgi, Pilpel & Oren (2010).

Regarding differential expression of REs in Figs. 3 and 4, ERVs, particularly the LTR class of HERV elements, are the most affected, as also occurs in psoriatic skin (Lättekivi et al., 2018) and osteosarcoma (Ho et al., 2017). A possible explanation may be that HERV elements (and, by extension, HERV-derived REs) are subject to strong epigenetic controls that keep them essentially silent, except in cancer, autoimmunity and placental development (Hurst & Magiorkinis, 2017). Another explanation may be based on a recent report (Kang, 2018) suggesting that tumour-initiating cells must have repressed HERV elements, since their expression produces different types of RNAs that elicit interferon response and apoptosis, leading to cancer cell death. On the other hand, HERV silencing enables cancer protection against lethal drug exposure, promotes immune evasion, and progresses to fully established cancer.

The fact that lung cancers display more down-regulation than up-regulation of HERVs and REs in general coincides with recent transcriptomic studies of REs in psoriatic skin (Lättekivi et al., 2018) and osteosarcoma (Ho et al., 2017), and may support the potential capability of REs not only as diagnostic biomarkers, but also as prognostic biomarkers for specific diseases, as reported in other studies (Ahn et al., 2013; Kabanov & Tishchenko, 2015). Since (i) RE expression seems to depend on read-through transcription (Gnanakkan et al., 2013; Deininger et al., 2017), (ii) RE expression change no longer seems to be related with class, type or even family (Fig. 5 and Tables S3 and S6), and (iii) it would be of future interest to determine whether RE-nearby areas are influencing their expression (mainly by demethylation or cross-transcription), as experimentally demonstrated for L1 in cell cultures (Deininger et al., 2017; Philippe et al., 2016), LCAL1 in lung cancer (White et al., 2014) and our recent prospective study in prostate cancer (Larrosa et al., 2018).

REs as potential lung cancer biomarkers

It is widely accepted that access to broad molecular screening as part of routine care will change the clinical management of lung cancer patients in the near future. The number of patients benefitting from targeted therapies needs an increase. The results presented here on the differential expression of REs in lung cancer confirm REs a new source of specific biomarkers for LUAD or SCLC, as anticipated in Gnanakkan et al. (2013).

Results in Fig. 7 confirm the potential use of three REs (AluYg6, LTR18B and MER126) as biomarkers to confirm at least two types of lung cancer. Although MER126 appeared to be a very attractive biomarker as it codes for the miRNA precursors of △mir-383 and △mir-3934 that have some prognostic capability (its down-regulation correlates with higher malignancy and a worse cancer prognosis (Ahn et al., 2013), it was finally excluded as a biomarker after validation in a new cohort of patients in which some cases do not present a neat MER126 down-regulation (Fig. 7B). Hence, only AluYg6 and LTR18B retained an intact potential biomarker capability in all patients that might be of use in future theranostic testings for lung cancer.

The biomarker capability of AluYg6 and LTR18B is also supported by some reports in the literature. AluYg6, a young Alu (SINE type) appearing only in human and chimpanzee genomes (Styles & Brookfield, 2007), presents several hundreds of copies but only very few are active as transposons (Salem et al., 2003). In line with results shown in Figs. 3, 4 and 7, the AluY family is usually up-regulated from 4-fold to 10-fold when camptothecin-treated tumour cells are compromised in apoptosis (Kabanov & Tishchenko, 2015). Therefore, we can speculate that down-regulation of AluYg6 might reflect the tumour status of a lung cell. Moreover, LTR18B could affect nearby gene expression due to the known cis-regulatory activity of LTRs belonging to ERVs (Gogvadze et al., 2009; Pavlicev et al., 2015; Trizzino et al., 2017; Trizzino, Kapusta & Brown, 2018). It seems then plausible that RE down-regulation could be more likely related to genomic environment and methylation than to REs per se.

The biomarker candidates AluYg6 and LTR18B cannot distinguish between SCLC and LUAD, but this issue can be overcome with HERVK11D-Int (bulleted in Table S2) as a LUAD-specific biomarker, and UCON88 (bulleted in Table S5) as a SCLC-specific biomarker for confirmatory diagnosis. HERVK11D-Int is also differentially expressed in psoriatic skin (Lättekivi et al., 2018) following a similar bioinformatic approach to the one described here. UCON88 is one of the ancient repeat sequences in RepBase that has not been classified (Kojima, 2018). Its name comes from the ‘UltraCONserved’ repeat between several species. It is tempting to link it with other well established ‘ultraconserved elements’ (UCEs) in human, mouse and rat genomes, which are now considered a new class of genomic regulatory elements that can play a critical role in human diseases such as cancer, even though their functions remain unknown (Baira et al., 2008). Even if we have not been able to find any similarity between UCON88 and UCEs, the potential association of UCON88 with gene expression changes cannot be ruled out (Jurka et al., 2012).

The scientific literature supports the biomarker capability of satellite REP522 in non-SCLC cell lines (Horie et al., 2017), leading to cancer-specific expression of some lncRNAs. Our study found that REP522 is not a biomarker candidate in spite of its specific up-regulation in SCLC cells. Moreover, since it is also expressed in LUAD cells from MRH patients (right Venn diagram of Fig. 8), it does not meet our proposed properties for a biomarker.

The consistent down-regulation of AluYg6 and LTR18B in all patients analysed, and the exclusive differential up-regulation of HERVK11D-Int and UCON88 in LUAD and SCLC cells, respectively, strengthen the idea that studies in the search for new biomarkers based on RE profiling merit the effort.

Conclusions

We have described an automated workflow using established bioinformatic tools to study expression change for REs in matched samples, revealing that their expression is clearly different in normal and tumour lung cells. Surprisingly, some REs present differential expression in normal cells from different cohorts, suggesting that patients, laboratories or protocols can affect expression quantification. Comparing expression values per patient for each RE, as performed here with logFCpP, seems to retain any expression difference due to an oncogenic change. Consistent expression change was revealed for 15 REs in LUAD patients and 71 in SCLC, with a bias to down-regulation of HERV-derived REs. A potential lung cancer capability was proposed for those REs that present the same expression change (up-regulation or down-regulation) in all analysed patients, rather than in most patients. AluYg6 and LTR18B down-regulations meet this property for both LUAD and SCLC, and were validated in a different LUAD-patient cohort. Additionally, HERVK11D-Int seems to be specifically up-regulated in LUAD (and not in SCLC), while UCON88 is up-regulated in SCLC (and not in LUAD). Even if RE expression results from a read-through transcription or from their own sequence, the data presented here indicate that some expression changes are consistent and disease-specific, but not patient-specific. Hence, studies focusing on the RE transcriptome merit the effort in cancer since REs seem a promising source of new, valuable lung cancer biomarkers. We also propose to extend this approach to other cancers and even other diseases.

Supplemental Information

Figure S1 Overall RE expression change per patient

For LUAD (upper panel) and SCLC (lower panel), the logFCpP (ordinate) of all REs in each patient (abscissa) are represented. logFC = 0 is marked with a green line to show that the overall level of RE expression can be considered unchanged in each patient after an oncogenic process.

Click here for additional data file.

Table S1 Name, class and family of differentially expressed REs when normal lung from SCLC is compared with normal lung from LUAD

Positive logFCOE values indicate up-regulation in normal cells of SCLC, while negative values indicate up-regulation in normal cells of LUAD samples.

Click here for additional data file.

Table S2 Name, class and family of differentially expressed REs in LUAD together with their logFCOE and their statistical significance as an FDR

These data were used in Fig. 3.

Click here for additional data file.

Table S3 Expression change by RE family and class in LUAD. The single statistically significant family is marked with an asterisk (*)

Click here for additional data file.

Table S4 Name, class and family of differentially down-regulated REs in SCLC together with their logFCOE and their statistical significance as an FDR

These data were used in Figs. 4 and 5.

Click here for additional data file.

Table S5 Name, class and family of differentially up-regulated REs in SCLC together with their logFCOE and their statistical significance as an FDR

These data were used in Figs. 4 and 5.

Click here for additional data file.

Table S6 Expression change by RE family and class in SCLC. The statistically significant families and class are marked with an asterisk (*)

Click here for additional data file.

This study would not have been possible without the computer resources and the technical support provided by the Andalusian Platform for Bioinformatics and the High-Throughput Sequencing Unit of the University of Malaga.

Additional Information and Declarations

Competing Interests

Author Contributions

Human Ethics

Data Availability

The authors declare there are no competing interests.

Macarena Arroyo performed the experiments, analyzed the data, prepared figures and/or tables, authored or reviewed drafts of the paper, approved the final draft.

Rocío Bautista conceived and designed the experiments, performed the experiments, analyzed the data, contributed reagents/materials/analysis tools, prepared figures and/or tables, authored or reviewed drafts of the paper, approved the final draft.

Rafael Larrosa performed the experiments, contributed reagents/materials/analysis tools, prepared figures and/or tables, authored or reviewed drafts of the paper, approved the final draft.

Manuel Ángel Cobo conceived and designed the experiments, analyzed the data, contributed reagents/materials/analysis tools, authored or reviewed drafts of the paper, approved the final draft.

M. Gonzalo Claros conceived and designed the experiments, analyzed the data, prepared figures and/or tables, authored or reviewed drafts of the paper, approved the final draft.

The following information was supplied relating to ethical approvals (i.e., approving body and any reference numbers):

The Ethics Committee of the Regional Hospital of Málaga called “Comité de Ética de la Investigación (CEI) Provincial de Málaga” (CIF Q-9150013-B) approved and consented the current study.

The appropriate consent forms were signed to gain the approval for using data from Bioprojects EGAS00001000334 and ERP001058.

The following information was supplied regarding data availability:

The sequence data is available as BioProject PRJNA563806, BioSample SAMN12688457.

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
