# Peer review of "Biomarker potential of repetitive-element transcriptome in lung cancer"

_PeerJ, doi:10.7717/peerj.8277_

## Round 0.1 · original submission · Major Revisions

· Academic Editor

Major Revisions

Please address all critiques of the reviewers and revise your manuscript accordingly.

Reviewer 1 ·

Basic reporting

In the article entitled “Biomarker potential of repetitive-element transcriptome in lung cancer”, Arroyo et al have attempted to identify potential REs as biomarkers for lung cancer.
I found the bioinformatics analysis pipeline well-conceived and the authors have highlighted the importance of RE transcriptome for potential biomarker search. I think the article is of value to the general cancer-diagnostics community and suitable for publication in PeerJ.
The authors might consider connecting the introductory paragraphs better to emphasize the motivation behind the present study. For example, can authors comment on if RE-transcriptome has been investigated for any other cancer type or if the REs identified in this study has been implicated in other cancer types?

Experimental design

No comments

Validity of the findings

No comments

Additional comments

No comments

·

Basic reporting

Overall, this submitted paper is a good follow-up of their previous publication based on bioinformatic analysis of the transposable elements (TEs) for lung cancer. Overall the paper submitted to PeerJ is well-written with sufficient intro about the background of the transposons, REs and its potential relationship with cancer. Results and discussion are also informative and well-written. The abstract is clear and can summarize what they have found in this paper.
New potential bio-markers for lung cancer treatment are very important and always can draw attention for biologists and pharmaceutical scientists. I think the authors have provided good results (figures and table, etc.) to support their claims, with appropriate references. I only have some concerns about the experimental design and data shown in this paper. I will describe these concerns in the following parts.
In general, I think I recommend this paper to be published in PeerJ if the authors can address all my concerns.

Experimental design

The bioinformatic analysis is well-designed with a very clear workflow.
My first question is about the data pre-processing stage. The authors claimed (lines 174-175) that “All snRNA, rRNA and tRNA elements were removed from output because they can provide spurious differences due to the way each laboratory extracted the RNAs.” I am curious that how much difference you will get by incorporating these three elements? It seems that the RepEnrich (BMC Genomics volume 15, Article number: 583 (2014)) method does include all the small RNAs.
My second question is related to the samples you collected from the database and this is just on my own curiosity. Can you distinguish on which stage the lung cancer cells were collected and RNA-sequenced? I am a little worried that the early stage cancer (Stage I) and late stage cancer (Stage II and III) may have differences in up- and down-regulations.

Validity of the findings

Overall, I think the paper have shown some interesting findings. My concern about the findings include:
1. In Table S1, the upregulation in SCLC and LUAD are expressed as positive and negative values. Usually the positive value represents the upregulation and negative represents the down regulation. It is a little confused that the negative values are used to illustrate the upregulation in LUAD. This is also not consistent with the positive and negative values used in the main figures.
2. In lines 219-220, the authors said, “only 5 were slightly (−1.063 to −1.39) up-regulated in normal cells from LUAD samples”. However, in line 231, log data in the same range (1.1-1.3, -1.1- -1.3) are named as “very significant fold changes”. These different claims are very confused. Appropriate explanations need to provide.
3. In Figures 3 and 4, the authors claimed that “all box medians are close to triangles”, which are not correct in some Res, such as HERVK11D−int and LTR18B in figure 3, ERV24B_Prim−int and MER65−int in Figure 4. Again, Appropriate explanations need to provide in the figure captions.
4. In lines 259-260, it seems in figure 4 there are only 21 Res are upregulated but 50 are downregulated, it is not consistent with the “46 differentially 260 expressed REs were down-regulated against 25 differentially expressed REs up-regulated” claim.
5. In figure 5, why is ‘DNA?’ so tightly distributed? Where is the red box?

Additional comments

In general, it is an interesting, timely paper that shows good results of potential biomarkers in lung cancer using bioinformatic analysis. I recommend this paper published in PeerJ if they address all my concerns listed above.

·

Basic reporting

no comment

Experimental design

no comment

Validity of the findings

no comment

Additional comments

1) In the introduction the authors emphasize that new bio-markers were required in the diagnosis of lung cancer and have rightfully explored the repetitive transposon elements that undergo oncogenic changes. However authors have not commented on the value of the novel bio-markers such as CBX3 and CRABP2 that have been similarly identified using biochemical methods and RNA-seq differential analysis of normal vs cancer tissue without using the global-RNA seq data. Can authors comment on the contrast of utilizing their method to the general transcriptome analysis (such as in Lung Cancer. Volume 84, Issue 3, June 2014, Pages 229-235) of the lung cancer and also explain if their method is superior to others and why or is their method just an alternative way of looking at more bio-markers?

2) Line 422, authors suggest the RE down-regulation could probably be due to methylation or genetic environment and are unsure how much does intrinsic RE transition contributes to the down-regulation. Can authors address this question in the current article? Specifically it is known that Alu/LINE-1 at specific gene-loci is an important information into the pathological states in cancer. It is not clear if authors have studied the locus-specific methylation of the transcriptome that they consider will be useful bio-marker of lung cancer. Do authors have a comment to dispel the ambiguity?

---

## Round 0.2 · accepted · Accept

· Academic Editor

Accept

Since all the critiques were adequately addressed, the manuscript is acceptable now.

·

Basic reporting

I think the revised manuscript is well-written and already addressed all my concerns. It is now suitable for publication in PeerJ and also can provide valuable information of potential biomarkers against the lung cancer.

Experimental design

I think the experimental design in current version is clear. I have no further comments.

Validity of the findings

The authors already addressed all the concerns I mentioned. I have no further comments.

·

Basic reporting

no comment

Experimental design

no comment

Validity of the findings

no comment